# Are Methylation Patterns in the *KALRN* Gene Associated with Cognitive and Depressive Symptoms? Findings from the Moli-sani Cohort

**DOI:** 10.3390/ijms251910317

**Published:** 2024-09-25

**Authors:** Miriam Shasa Quiccione, Alfonsina Tirozzi, Giulia Cassioli, Martina Morelli, Simona Costanzo, Antonietta Pepe, Francesca Bracone, Sara Magnacca, Chiara Cerletti, Danilo Licastro, Augusto Di Castelnuovo, Maria Benedetta Donati, Giovanni de Gaetano, Licia Iacoviello, Alessandro Gialluisi

**Affiliations:** 1Department of Epidemiology and Prevention, IRCCS Neuromed, Via dell’Elettronica, 86077 Pozzilli, Italy; quiccioneshasa@gmail.com (M.S.Q.); nina.tirozzi@gmail.com (A.T.); martina-morelli2009@libero.it (M.M.); simona.costanzo@moli-sani.org (S.C.); pepe.antonietta@gmail.com (A.P.); francesca.bracone@moli-sani.org (F.B.); sara.magnacca@moli-sani.org (S.M.); chiara.cerletti@moli-sani.org (C.C.); dicastel@moli-sani.org (A.D.C.); mbdonati@moli-sani.org (M.B.D.); giovanni.degaetano@moli-sani.org (G.d.G.); alessandro.gialluisi@moli-sani.org (A.G.); 2Department of Experimental and Clinical Medicine, University of Florence, 50134 Florence, Italy; cassioli.giulia@gmail.com; 3Area Science Park, 34149 Trieste, Italy; danilo.licastro@areasciencepark.it; 4Department of Medicine and Surgery, LUM University, 70010 Casamassima, Italy

**Keywords:** kalirin, *KALRN*, cognitive performance, depressive symptoms, MoCA, PHQ9

## Abstract

The *KALRN* gene (encoding kalirin) has been implicated in several neuropsychiatric and neurodegenerative disorders. However, genetic evidence supporting this implication is limited and targeted epigenetic analyses are lacking. Here, we tested associations between epigenetic variation in *KALRN* and interindividual variation in depressive symptoms (PHQ9) and cognitive (MoCA) performance, in an Italian population cohort (N = 2409; mean (SD) age: 67 (9) years; 55% women). First, we analyzed the candidate region chr3:124584826–124584886 (hg38), within the *KALRN* promoter, through pyrosequencing of 1385 samples. Then, we widened the investigated region by analyzing 137 CpGs annotated to the whole gene, rescued from epigenome-wide (Illumina EPIC) data from 1024 independent samples from the same cohort. These were tested through stepwise regression models adjusted for age, sex, circulating leukocytes fractions, education, prevalent health conditions and lifestyles. We observed no statistically significant associations with methylation levels in the three CpGs tested through pyrosequencing, or in the gene-wide association analysis with MoCA score. However, we observed a statistically significant association between PHQ9 and cg13549966 (chr3:124106738; β (Standard Error) = 0.28 (0.08), Bonferroni-corrected *p* = 0.025), located close to the transcription start site of the gene. This association was driven by a polychoric factor tagging somatic depressive symptoms (β (SE) = 0.127 (0.064), *p* = 0.048). This evidence underscores the importance of studying epigenetic variation within the *KALRN* gene and the role that it may play in brain diseases, particularly in atypical depression, which is often characterized by somatic symptoms.

## 1. Introduction

*KALRN* (*kalirin RhoGEF kinase*), also known as ARHGEF24, is a 66-exon gene located on chr 3q21 and encoding Kalirin [1], a member of the Rho-guanosine (GEF) family. Several isoforms of Kalirin are generated by alternative splicing, the most abundant isoforms in the central nervous system (CNS) being Kalirin-7, -9, and -12 [2]. Each isoform provides a specific contribution to neurite and dendritic outgrowth and to dendritic arborization [3].

Both genetic and molecular studies highlight the importance of normal *KALRN* expression for healthy neurodevelopment and function [1]. Recent studies demonstrated that Kalirin-7 physically and functionally interacts with several proteins previously implicated in schizophrenia like DISC1 (Disrupted-in-Schizophrenia-1) [4], and that its levels are reduced in the prefrontal cortex [5]. *KALRN* was also linked with schizophrenia risk through genetic analyses of rare variants—along with depression [6]—and through postmortem analyses of cortical *KALRN* mRNA and protein levels within individuals with autism [5].

Moreover, *KALRN* was associated with different neurodegenerative disorders due to the accumulation of neurotoxic aggregates, including Huntington’s disease (HD) [7], Alzheimer’s disease (AD) [8] and Parkinsons’s Disease (PD) [9]. Since a reduction in cortical Kalirin-7 in HD mice is paralleled by early cortical dendritic spine alteration and cognitive defects, it has been hypothesized that the cortical function could be restored by increasing the levels of Kalirin-7 [10].

Recent studies have confirmed that since Kalirin-7 is a key regulator of dendritic spine morphogenesis and maintenance in forebrain pyramidal neurons [11], changes in its expression, mutations, or alterations in its upstream or downstream signaling partners lead to an aberrant dendritic spine number and morphology [5]. This happens, for instance, in AD [10], where Amyloid-β (Aβ) may directly lead to Kalirin-7 dysregulation or loss [3]. This induces an immature dendritic spine phenotype, which can be prevented by overexpression of Kalirin-7 [11]. Since AD patients with psychoses display an increase in beta-amyloid (Aβ42/40) ratio as well as decreased Kalirin levels, loss of *KALRN* expression has been correlated with AD pathology [1], while Kalirin-7 pharmacological activation may be protective in AD, preventing or delaying synapse pathology [11]. Recent studies have also suggested a potential implication of *KALRN* in the pathogenesis of PD, either through genetic evidence [9,12,13] or through indirect evidence of an association between Kalirin-7 and Synphilin-1, an interaction partner of α-synuclein [14]. Indeed, Kalirin-7 promotes the recruitment of synphilin-1 inclusions into aggresomes (organelles that serve as storage compartments for misfolded proteins) and increases the susceptibility of synphilin-1 inclusions to be degraded [15].

Despite these findings suggesting an implication of *KARLN* in many neurodegenerative disorders and related neuropsychiatric phenotypes, to our knowledge, no study has tested associations between epigenetic variation in the gene—which may affect its function and expression levels—and interindividual variation in depressive symptoms and cognitive performance. To fill in this knowledge gap, we tested this relation in an Italian population cohort.

## 2. Results

The two analyzed Moli-sani sub-cohorts showed differences in sociodemographic variables (Appendix A), which are reasonable given the older age of the participants analyzed through the EPIC array, in a study originally aimed at clarifying the effects of polypharmacy in elderly.

Indeed, the subjects analyzed through pyrosequencing were more prevalently women (58,0%; *p* < 0.002), younger (mean (SD) age: 63.11 (9.07) vs. 71.21 (6.47) years; *p* < 0.001) and more educated (*p* < 0.001), compared to subjects analyzed through the EPIC array. However, no significant differences were observed between the two populations regarding the prevalence of the main health conditions (CVD, diabetes, dyslipidemia, cancer) and BMI distribution. The pyrosequenced sub-cohort also showed a higher prevalence of current smokers (17.3 vs. 14.0%), a lower alcohol intake (9.5 (14.5) vs. 13.6 (18.4) g/d) and a slightly higher cognitive performance (MoCA score 24.8 (3.3) vs. 23.4 (4.4); *p* < 0.001).

In the sub-cohort undergoing targeted pyrosequencing of the candidate region chr3:124584826–124584886 (Figure 1), none of the multivariable models testing MoCA and PHQ9 scores revealed significant associations with methylation levels in the CpGs tested, neither when testing the three CpGs separately (Table 1) nor when testing the three CpGs together in a single model (Appendix A).

In the sub-cohort undergoing methylation analysis through the EPIC array, 25 and 23 CpGs annotated to *KALRN* were retained in the linear models predicting MoCA and PHQ9 scores, respectively (Table 2a,b). The most significant association observed with the MoCA score was detected at cg10103145 (chr3:124488112; β (SE) = −0.18 (0.06); raw *p*-value = 0.0016), which, however, did not survive Bonferroni correction for multiple testing (corrected *p* = 0.08; Table 2a). On the contrary, we observed a significant association surviving Bonferroni correction for multiple testing with PHQ9, for cg13549966 (chr3:124106738; β (SE) = 0.28 (0.08), raw *p*-value = 0.0005, adjusted *p*-value = 0.025; Table 2b).

The polychoric factor analysis of PHQ9 items allowed us to extract two factors tagging cognitive and somatic depressive symptoms (Appendix A), which were highly correlated (Pearson’s r = 0.81). One of them (hereafter called MR1) showed high loadings of cognitive symptoms and the other one (hereafter called MR2) showed high loadings of somatic symptoms. These factors explained 27% and 30% of the common variance shared among the depressive symptoms tested, respectively. In multivariable regression modeling, including each factor vs. cg13549966 methylation levels (Appendix A), we observed a significant association of the somatic factor (most conservatively adjusted model: β (SE) = 0.127 (0.064), *p* = 0.048), which was stable across all the models incrementally adjusted as above and was independent on the cognitive factor (used as covariate). Conversely, the cognitive factor did not show any association with cg13549966 methylation levels, in any of the models tested (Appendix A). Similarly, single-PHQ9 items did not show any association in multivariable logistic regression models adjusted for all the covariates mentioned above and for all the items other than the outcome (Appendix A).

## 3. Discussion

In this paper, we tested the association between cognitive and psychiatric decline—assessed through MoCA and PHQ9 scores, respectively—and the epigenetic variability of the *KALRN* gene (chr3q21.2). In the first instance, we tested three CpGs in the candidate region chr3:124584826–124584886, which hosts the promoter of a brain-specific isoform of the gene encoding Kalirin-2. This did not reveal any association with MoCA and PHQ9 scores, neither when tested separately nor when tested jointly in multivariable models. However, when we widened the analyses to 137 CpGs annotated to the *KALRN* gene, rescued from an independent sub-cohort of the Moli-sani study, this revealed a suggestively significant association of cognitive performance with cg10103145—a CpG located close to an eQTL of the gene [16] in the brain (especially frontal) cortex, a few bp from the junction between intron 28 and exon 29—and a statistically significant association of depressive symptoms with cg13549966—located within the intron 1 of many isoforms, close to the transcription start site and within a putative Cis-Regulatory Element (CRE) of the gene. Further analyses revealed that the significant association of cg13549966 was mainly driven by a polychoric factor tagging somatic depressive symptoms, while the cognitive symptoms factor did not show any association. This evidence suggests that altered epigenetic patterns in this site may be related mostly to neurovegetative modifications linked to depression, like altered sleeping, appetite/eating, low energy and all those symptoms most strictly related to sickness behavior, which characterize atypical—and often treatment-resistant—depression [17]. Indeed, a recent cluster analysis of major depression cases from China revealed the existence of two clusters—characterized by affective and somatic symptoms, respectively—with the somatic cluster being significantly associated with a poorer treatment efficacy and prognosis [18]. Therefore, *KALRN* may represent a potential candidate for both clinical and experimental studies on somatic and treatment-resistant depression.

While it was not possible to meta-analyze the two studies due to the non-overlapping CpGs tested, we followed up these top hits through a search on common Epigenome Wide Association Scan (EWAS) databases, namely EWASdb [19] and EWAS Catalog [20], but we did not find any significant association reported.

A deeper search of the literature revealed that cg13549966 was also previously associated with airflow obstruction—measured as a ratio between forced expiratory volume in one second (FEV1) and forced vital capacity (FVC) below the lower limit of normal—in people living with Human Immunodeficiency Virus infection [21].

Although we were not able to find a study reporting differences in kalirin expression between depression cases and controls, we found evidence of a decrease in Kalirin-7 levels in the hippocampus of male rats exposed to chronic social defeat stress, one of the triggering mechanisms used to model major depression in rodents [22]. Similarly, chronic unpredictable mild stress, used to recapitulate depression-like behaviors in rodents and causing a decrease in spine density in the orbitofrontal cortex, was accompanied by both depression-like behaviors and decreased expression of kalirin-7 in this region [23]. Evidence of under-expression of *KALRN* in comorbid conditions like schizophrenia was also reported, both at the mRNA [24] and at the protein expression level in postmortem brains of patients [25,26]. Similar evidence was found for another comorbid condition, Alzheimer’s disease [27,28].

Over the years, other studies have investigated epigenetic variations associated with the *KALRN* gene and its role in various diseases. Rizzo et al. [29] conducted an EWAS and a regional analysis in a large Hispanic population, reporting that maternal obesity or diabetes mellitus during pregnancy was associated with an infant hypomethylation of a CpG island located ~120 kb far from the region analyzed here (cg20807374). More recently, Poisel et al. [30] performed an EWAS of cocaine use disorder patients, in human brain samples. This analysis showed that the *KALRN* gene harbors a differentially methylated region located at chr3:124584681–124585247, finding direct links to a potential role in the biological response to cocaine. The results of these studies, along with the suggestive evidence observed here, underscore the importance of continuing to study the epigenetic variations associated with the *KALRN* gene and the role that they may play in human—especially brain—diseases, possibly in larger, epigenome-wide samples.

### Strengths and Limitations

This preliminary study is novel since it represents the first attempt to test the association between specific psychiatric and cognitive functions and the epigenetic variability in the *KALRN* gene, in a relatively large population cohort, providing a contribution to the field of neurogenetics and to the epigenetic epidemiology of cognitive and mood decline.

However, our study also presents some limitations. First is the lack of functional validation of the statistical evidence reported here, e.g., testing whether the identified CpG (cg13549966) is associated with changes in *KALRN* expression, which we plan to carry out in the future, along with other functional validation analyses. Second is the use of two different methods to test epigenetic variation in the *KALRN* gene, which may represent a differential confounding factor between the analyzed sub-cohorts. While we initially focused on pyrosequencing analysis to have a more fine-grained picture of a small candidate region with a likely regulatory role, the resulting region analyzed was very small, due to technical limitations of the instrument used. The cost of the analyses and the total lack of even suggestive associations in pyrosequencing analyses prompted us to stop this kind of investigation and focus our attention on association scans at the entire gene level—exploiting array data which became available in the meantime. This provided larger statistical power and revealed interesting associations, which, however, were not located in the same target region analyzed in the first sub-cohort. This made it unfeasible to meta-analyze the two sub-cohorts and to boost the power of our analyses. Still, EPIC array analyses of the pyrosequenced samples are now planned. This will help to analyze all the samples in a homogeneous way and to enlarge the meta-analysis—possibly also with other independent cohorts that may become available in the meantime—and will help clarify potential links between *KALRN* epigenetic patterns and behavioral traits.

## 4. Materials and Methods

### 4.1. Subjects and Samples

The Moli-sani study is a prospective cohort study established in 2005–2010 with the enrolment of 24,325 men and women (aged ≥ 35 y) randomly recruited from the general population of Molise, a Southern Mediterranean Italian region, to investigate risk factors in the onset of cardiovascular, cerebrovascular and cancer diseases. Details of the study are available elsewhere [31,32]. A sub-cohort of 2581 participants was recalled and fully re-examined in 2017–2020 [33]. Among these, we analyzed 1474 samples, stored in the Neuromed Biobanking Center, to investigate differentially methylated regions of the *KALRN* gene associated with psychiatric and cognitive performances, through the pyrosequencing of a candidate region in the promoter region of brain-specific isoform 2 (Gencode Transcript: ENST00000682363.1), where methylation patterns may be more likely to have a potential regulatory role in the expression of the gene. In total, 1099 additional subjects who were undergoing methylation analyses in the context of an epigenome-wide association scan (EWAS) were involved in the analysis in a second stage.

DNA pyrosequencing of the region chr3:124584826–124584886 was conducted as described in detail in the Appendix A (see the *DNA pyrosequencing and quality control* paragraph). Briefly, after DNA extraction from buffy coats through a salting-out in-house protocol and bisulfitation via bisulfite treatment of DNA using EZ-96 DNA Methylation Kit (ZYMO RESEARCH^®^, Irvine, CA, USA), we performed the pyrosequencing reaction using the PyroMark Q48 instrument (Tegelen, The Netherlands), according to the manufacturer’s instructions. Through specific sequencing primers, we analyzed the region chr3:124584826–124584886 (hg38), identifying three different CpGs, located in chr3:124584830, chr3:124584833 and chr3:124584849-C-G, also known as rs56407180.

In total, 1099 samples analyzed through the Illumina EPIC array (v1, 865,918 CpGs)—in the context of a project investigating epigenetic changes associated with polypharmacy in elderly—were extracted and underwent bisulfite treatment as above, while QC of methylation signals was carried out in ChAMP v2.20.1 [34]. Probes were filtered out if they showed a detection of *p* > 0.01, i.e., fewer than 3 beads in more than 5% of all samples; if they were cross-reactive probes; if they were located on sex chromosomes; or if they showed a biased DNA methylation signal due to Single-Nucleotide Polymorphisms detected at CpG sites. Computed beta values underwent normalization, identification and removal of batch effects through singular value decomposition (svd) and combat analysis. From the resulting (668,413) probes passing QC, we rescued 137 candidate CpGs out of 165 total probes annotated to the *KALRN* gene, based on the EPIC BeadChip manifest (Appendix A), to analyze the association with methylation patterns across the whole gene and not only in a targeted region. Full details of this analysis are reported in the Appendix A (see the *DNA methylation analysis through Illumina EPIC array* paragraph).

### 4.2. Cognitive and Psychiatric Assessment

Depressive symptoms were assessed via the Patient Health Questionnaire 9 (PHQ9), a common tool for depression screening in primary care [35]. The PHQ9 scale assesses the severity of depression by scoring the frequency of nine specific domains, which typically affect depressed individuals, in the two weeks preceding the interview. These include anhedonia, low mood, altered sleep patterns or eating behaviors, feelings of failure/low self-esteem, fatigue, mental concentration problems, hypo/hyperactive behaviors and suicidal ideation [36]. As a severity measure, the PHQ9 score can range from 0 to 27, since each of the 9 items can be scored from 0 (not at all) to 3 (nearly every day) [35].

Cognitive performance was assessed through The Montreal Cognitive Assessment (MoCA), a measure of global cognitive function [37]. It was originally developed to detect mild cognitive impairment (MCI) but is now often used as a screening tool for dementias [38]. This evaluates short-term memory, visuospatial function, executive function, attention, concentration and working memory, language and orientation [39]. The raw score is adjusted for education level (1 extra point for 10–12 years of formal education; 2 points added for 4–9 years of formal education). The maximum score is 30 points: normal cognitive function is indicated by a MoCA score between 26 and 30 points [37]; a score between 25 and 18 indicates cognitive impairment, while MoCA ≤ 18 suggests the presence of a form of dementia.

### 4.3. Statistical Analysis

All statistical analyses were carried out in R v 4.0.4 [40] or Jamovi (Version 2.3.28 for macOS) [41].

#### 4.3.1. Analysis of Pyrosequencing Data

Among 1474 unique samples that underwent targeted pyrosequencing, subjects were removed when they showed an unreliably low MoCA (≤10, n = 13) or missing MoCA/PHQ9 score (n = 28), or unreliable medical or dietary questionnaires (n = 2). Individuals with an education level not declared (n = 2) or with a missing methylation measure for at least one of the CpGs analyzed were also removed. Other missing data were imputed through a k-nearest neighbor algorithm of the VIM package (*kNN()* function, k = 10) [42] to rule out potential bias from missing-not-at-random (MNAR) data patterns. Distributions of methylation levels for each CpG analyzed are reported in Appendix A. To ensure robustness against potential outliers in methylation levels, we computed the log2 of beta values for each CpG. Values resulting in log2(0) were treated as missing data (6 for CpG1, 3 for CpG2, 10 for CpG3). This left, for subsequent analyses, 1385 samples with complete phenotypic and epigenotype information available.

To model the relationship between cognitive performance (MoCA score) and epigenetic variability in the analyzed candidate region, we built multivariable generalized linear models incrementally adjusted for (i) age, sex and granulocytes, monocytes and lymphocytes fractions, to account for the potential confounding of white blood cell populations heterogeneity (Model 1); (ii) education level completed (Model 2); (iii) prevalent health conditions such as cardiovascular disease (CVD), diabetes, dyslipidemia, cancer, and body mass index (BMI) (Model 3); (iv) lifestyles or their proxies like lifestyle (LIS) and dietary inflammatory score (DIS, Appendix A), adherence score to Mediterranean Diet (MD), alcohol (g/day) and energy intake (Kcal/day), and smoking status (Model 4); and finally, (v) depressive symptoms (PHQ9) (Model 5). The same approach was used to model the relationship between depressive symptoms (PHQ9) and the candidate CpGs mentioned above, except for the most enriched model, which was adjusted for MoCA score. For both outcomes, multivariable models were first built for each candidate CpG separately, and then jointly with all CpGs together, including all the covariates. All the covariates used in the analysis are explained in the Appendix A. A Bonferroni correction for multiple testing was applied as appropriate, based on the number of CpGs and traits analyzed. Specifically, we corrected the α threshold for three CpGs and two traits tested in the analysis of pyrosequencing data (for a total of six independent tests, α = 0.008).

#### 4.3.2. Analysis of Array Data

To analyze associations with MoCA and PHQ9 among the 1099 participants with CpG methylation levels measured throughout the *KALRN* gene, we applied the same QC filters mentioned above to data rescued from the EPIC array. Briefly, we filtered out subjects with an unreliably low MoCA score (≤10, n = 33), a missing MoCA/PHQ9 score (n = 36), unreliable medical or dietary questionnaires (n = 4), or an education level not declared (n = 2). No samples with missing methylation measures were detected, while other missing data were imputed through a k-nearest neighbor algorithm, as above [42]. Overall, 1024 samples were left for subsequent analysis. We carried out stepwise regression models through the *stepAIC()* function of the MASS package [43] with (default) “both” options, in order to keep within each model only those CpGs representing a gain in the tradeoff between parsimony and goodness of fit of the regression models, and allowing us to “clean” the models from potential collinearity bias among the different CpGs tested. In this case, only full generalized linear models were implemented (Model 5)—for both MoCA and PHQ9 score—setting as fixed (non-modifiable) covariates age, sex, education level completed, prevalent health conditions (CVD, diabetes, dyslipidemia, cancer and BMI), lifestyles or their proxies (LIS and DIS, alcohol and energy intake, smoking status, adherence score to MD).

To investigate in detail the associations found between depressive symptoms and *KALRN* epigenetic variants (see below), we conducted two complementary analyses. First, we extracted polychoric factors tagging cognitive and somatic depressive symptoms from the nine PHQ9 items assessed, as in [36], to test whether any of these two domains of depression was driving the association detected with the overall PHQ9 scale. These factors were modeled as outcomes in multivariable linear regression models, incrementally adjusted for age, sex and granulocytes, monocyte and lymphocyte fractions, education level, prevalent health conditions, lifestyles and MoCA score, and for a polychoric factor other than the outcome, to identify specific associations with each of the depressive domains identified. Second, to further deepen associations at the single-symptom level, we dichotomized each single item of the PHQ9 scale and tested it for association with the candidate CpG tested, in multivariable fully adjusted models, including all the covariates used above and all the depressive symptoms other than the one modeled as an outcome (see Appendix A for details).

In this analysis, we applied a Bonferroni correction for 48 independent tests (corrected α = 0.001), namely the sum of CpGs retained in the stepwise regression analysis of the MoCA (23 CpGs) and PHQ9 score (25 CpGs).

## 5. Conclusions

This study revealed a novel association between the methylation level of cg13549966, a CpG located in a regulatory region of the *KALRN* gene, and interindividual variation in depressive symptoms, in an Italian population cohort. This association was mainly driven by a factor representing somatic/neurovegetative symptoms, suggesting a more prominent implication of *KALRN* in somatic forms of depression, which is often characterized by treatment resistance. Therefore, Kalirin may represent a potential candidate for both clinical and experimental investigations on somatic and treatment-resistant depression, which warrants further large-scale genetic, epigenomic, and functional studies to validate these findings.

## Figures and Tables

**Figure 1 ijms-25-10317-f001:**
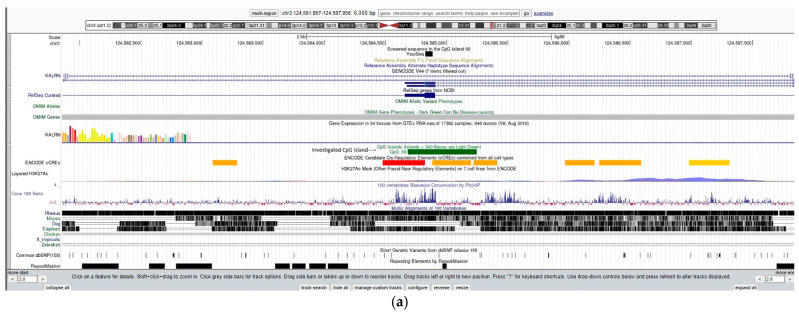
Candidate region analyzed through pyrosequencing. The analyzed chr3:124584826–124584886 (hg38) region—highlighted in green—is reported with (**a**) a 2 kb and (**b**) a 200 bp zoom. The picture was created through the UCSC Genome Browser tool (https://genome-euro.ucsc.edu, accessed on 22 January 2024).

**Table 1 ijms-25-10317-t001:** Results of the analysis of (**a**) cognitive performance (MoCA) and (**b**) depressive symptoms (PHQ9) vs. methylation levels of the three candidate CpGs tested.

**(a)**			
**Regression Model**	**CpG1** **β (SE), *p***	**CpG2** **β (SE), *p***	**CpG3** **β (SE), *p***
Model 1	−0.050 (0.163), 0.76	−0.084 (0.160), 0.60	0.016 (0.136), 0.91
Model 2	0.064 (0.151), 0.67	0.003 (0.149), 0.99	0.069 (0.127), 0.59
Model 3	0.065 (0.151), 0.67	0.011 (0.150), 0.94	0.073 (0.127), 0.56
Model 4	0.068 (0.151), 0.65	0.011(0.149), 0.94	0.079 (0.127), 0.53
Model 5	0.072 (0.151), 0.64	0.018 (0.149), 0.91	0.085 (0.127), 0.50
**(b)**			
**Regression Model**	**CpG1** **β (SE), *p***	**CpG2** **β (SE), *p***	**CpG3** **β (SE), *p***
Model 1	0.209 (0.216), 0.33	0.357 (0.212), 0.09	0.191 (0.181), 0.29
Model 2	0.206 (0.216), 0.34	0.359 (0.213), 0.09	0.207 (0.182), 0.25
Model 3	0.206 (0.215), 0.34	0.338 (0.213), 0.11	0.210 (0.181), 0.25
Model 4	0.165 (0.215), 0.44	0.313 (0.212), 0.14	0.203 (0.181), 0.26
Model 5	0.169 (0.215), 0.43	0.313 (0.212), 0.14	0.207 (0.181), 0.25

Association betas and relevant standard errors (β (SEs)) of the incrementally adjusted regression models are reported, along with (raw) *p*-values. Legend: Model 1: age + sex + lymphocytes (%) + monocytes (%) + granulocytes (%); Model 2: Model 1 + education level; Model 3: Model 2 + prevalent health conditions (cvd, diabetes, dyslipidemia, cancer, BMI); Model 4: Model3 + lifestyles (LIS, DIS and MD scores, alcohol (g/day) and energy intake (Kcal/day)); Model 5: Model 4 + MoCA or PHQ9.

**Table 2 ijms-25-10317-t002:** Results of the analysis of (**a**) cognitive performance (MoCA) and (**b**) depressive symptoms (PHQ9) vs. methylation levels of 137 CpGs tested in the *KALRN* gene.

**(a)**				
**CpG**	**Position (GRCh38)**	**β (SE)**	**Raw *p*-Values**	**Bonferroni *p*-Values**
cg10103145	124488112	−0.18 (0.06)	0.0016	0.08
cg05720721	124084281	−0.25 (0.08)	0.0023	0.11
cg22376833	124269069	0.19 (0.07)	0.0042	0.20
cg05633656	124197837	0.17 (0.07)	0.011	0.53
cg11148677	124157868	0.12 (0.05)	0.013	0.64
cg11688731	124657745	0.16 (0.06)	0.013	0.64
cg00086941	124315253	0.16 (0.07)	0.021	1
cg21274175	124659599	−0.15 (0.07)	0.024	1
cg00259083	124298882	0.22 (0.1)	0.03	1
cg16395286	124674322	−0.12 (0.05)	0.03	1
cg23475018	124558791	0.19 (0.09)	0.03	1
cg16100687	124347786	0.14 (0.06)	0.031	1
cg24323597	124094657	−0.13 (0.06)	0.039	1
cg12144803	124094536	−0.1 (0.05)	0.043	1
cg24175649	124585314	0.07 (0.04)	0.05	1
cg01734829	124621210	0.09 (0.05)	0.05	1
cg04807106	124384713	−0.15 (0.08)	0.06	1
cg07483657	124468583	0.09 (0.05)	0.06	1
cg07600936	124504032	−0.06 (0.03)	0.07	1
cg19918027	124268798	−0.1 (0.05)	0.07	1
cg24252694	124351590	−0.07 (0.04)	0.08	1
cg20570458	124351086	−0.09 (0.05)	0.11	1
cg23981549	124583877	−0.07 (0.05)	0.16	1
**(b)**				
**CpG**	**Position (GRCh38)**	**β (SE)**	**Raw *p*-Values**	**Bonferroni *p*-Values**
**cg13549966**	**124106738**	**0.28 (0.08)**	**0.0005**	**0.025**
cg26510634	124158957	−0.21 (0.07)	0.0053	0.25
cg18981420	124584694	0.19 (0.07)	0.0068	0.33
cg23837547	124584561	−0.18 (0.07)	0.0071	0.34
cg09990141	124563137	0.21 (0.08)	0.010	0.50
cg21167201	124559065	0.13 (0.06)	0.016	0.76
cg01718380	124266674	0.21 (0.09)	0.016	0.77
cg24323597	124094657	−0.16 (0.07)	0.024	1
cg07600936	124504032	−0.09 (0.04)	0.028	1
cg09751451	124521183	0.1 (0.05)	0.048	1
cg16919675	124518463	−0.1 (0.05)	0.05	1
cg24829557	124094618	0.17 (0.09)	0.06	1
cg02754399	124584717	−0.13 (0.07)	0.06	1
cg10908929	124208194	−0.15 (0.08)	0.06	1
cg12777296	124624268	−0.1 (0.05)	0.06	1
cg23440058	124384849	−0.13 (0.07)	0.07	1
cg24175649	124585314	−0.09 (0.05)	0.09	1
cg21618455	124231171	0.15 (0.09)	0.09	1
cg13779109	124407647	0.08 (0.05)	0.1	1
cg04516112	124584557	0.11 (0.07)	0.1	1
cg16100687	124347786	−0.12 (0.07)	0.1	1
cg16395286	124674322	−0.1 (0.06)	0.11	1
cg16278716	124094740	0.1 (0.06)	0.11	1
cg11148677	124157868	−0.08 (0.06)	0.13	1
cg10430690	124094749	0.08 (0.06)	0.15	1

*p*-values were rounded to the fourth decimal place if <0.01, to the third decimal place if <0.05, and to the second decimal place otherwise. Associations surviving Bonferroni correction for multiple testing (α = 0.001) are highlighted in bold. Legend: β (SE): Association beta and relevant standard errors.

## Data Availability

The data underlying this article will be shared upon reasonable request to the corresponding author. The data are stored in an institutional repository (https://repository.neuromed.it) and access is restricted by the ethical approvals and the legislation of the European Union.

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
