# Peer review of "Are Methylation Patterns in the KALRN Gene Associated with Cognitive and Depressive Symptoms? Findings from the Moli-sani Cohort"

_ijms, 2024, doi:10.3390/ijms251910317_

Round 1

Reviewer 1 Report

Comments and Suggestions for Authors

Recent studies suggest that KALRN (kalirin RhoGEF kinase) gene can play a role in the pathogenesis of different neuropsychiatric disorders from depression to Alzheimer’s disease, although no studies have been conducted to analyze the epigenetics of KALRN in depressive disorders and its impact on cognitive dysfunction and affective and somatric symptoms in these diseases..

The manuscript entitled “Are methylation patterns in the KALRN gene associated with 2 cognitive and depressive symptoms? Findings from the Moli-3 sani cohortexamines the  associations between epigenetic variation 22 in KALRN and interindividual variation in depressive symptoms (PHQ9) and cognitive (MoCA) 23 performance, in an Italian population cohort (N=2,409; mean (SD) age: 67 (9) years; 55% women).  By using  the gene-wide analysis, the authors demonstrate an association approaching significance for MoCA score with cg10103145 32 (chr3:124488112; β (SE) = -0.17 (0.056), corrected p-value = 0.056), and a significant association between PHQ9 and cg13549966 (chr3:124106738; 0.28 (0.078), p = 0.009), located close 34 to the transcription start site of the gene. Interestingtly the authors found for the first time a significant association of cg13549966 with a polychoric factor tagging somatic depressive symptoms. These results are clinically relevant because somatic symptoms can significantly contribute to treatment resistance in depression.

The article appears comprehensive and well-organized. The introduction is appropriate with the topic of the study. The study itself is interesting for its relevance and novelty. In summary, I recommend this manuscript to be considered for publication with only minor revision, as outlined below:

·       In the discussion section the authors might briefly discuss at line 306 how somatic symptoms are strongly associed to treatment-resistance in depression (see the recent paper from Zhou J, Zhou J, Feng Y, Feng L, Xiao L, Chen X, Feng Z, Yang J, Wang G. The novel subtype of major depressive disorder characterized by somatic symptoms is associated with poor treatment efficacy and prognosis: A data-driven cluster analysis of a prospective cohort in China. J Affect Disord. 2024 Feb 15;347:576-583. doi: 10.1016/j.jad.2023.12.005).

·       Please correct in the abstract “promotor” with “promoter”.

Author Response

Dear Editor,

Thank you and both the reviewers for considering our manuscript for publication on the IJMS. Please find below response to reviewer’s comments (in violet).

To highlight the changes throughout the manuscript, we uploaded a tracked changes version of the manuscript in pdf format.

Should you or any of the reviewers need any additional clarification, please do not hesitate to contact us.

Regards,

Alessandro Gialluisi (on behalf of all co-authors)

Reviewer #1:

Recent studies suggest that KALRN (kalirin RhoGEF kinase) gene can play a role in the pathogenesis of different neuropsychiatric disorders from depression to Alzheimer’s disease, although no studies have been conducted to analyze the epigenetics of KALRN in depressive disorders and its impact on cognitive dysfunction and affective and somatic symptoms in these diseases.

The manuscript entitled “Are methylation patterns in the KALRN gene associated with cognitive and depressive symptoms? Findings from the Moli-sani cohort” examines the associations between epigenetic variation in KALRN and interindividual variation in depressive symptoms (PHQ9) and cognitive (MoCA) performance, in an Italian population cohort (N=2,409; mean (SD) age: 67 (9) years; 55% women).  By using the gene-wide analysis, the authors demonstrate an association approaching significance for MoCA score with cg10103145 (chr3:124488112; β (SE) = -0.17 (0.056), corrected p-value = 0.056), and a significant association between PHQ9 and cg13549966 (chr3:124106738; 0.28 (0.078), p = 0.009), located close to the transcription start site of the gene. Interestingly the authors found for the first time a significant association of cg13549966 with a polychoric factor tagging somatic depressive symptoms. These results are clinically relevant because somatic symptoms can significantly contribute to treatment resistance in depression.

The article appears comprehensive and well-organized. The introduction is appropriate with the topic of the study. The study itself is interesting for its relevance and novelty. In summary, I recommend this manuscript to be considered for publication with only minor revision, as outlined below:

  • In the discussion section the authors might briefly discuss at line 306 how somatic symptoms are strongly associated to treatment-resistance in depression (see the recent paper from Zhou J, Zhou J, Feng Y, Feng L, Xiao L, Chen X, Feng Z, Yang J, Wang G. The novel subtype of major depressive disorder characterized by somatic symptoms is associated with poor treatment efficacy and prognosis: A data-driven cluster analysis of a prospective cohort in China. J Affect Disord. 2024 Feb 15;347:576-583. doi: 10.1016/j.jad.2023.12.005).

We thank the reviewer for its insightful suggestion. We added the following statement to the relevant Discussion paragraph:

“Indeed, a recent cluster analysis of major depression cases from China revealed the ex-istence of two clusters - characterized by affective and somatic symptoms, respectively – with the somatic cluster being significantly associated with a poorer treatment efficacy and prognosis [31].”

  • Please correct in the abstract “promotor” with “promoter”. For research article

Agreed. We changed the term as suggested.

Reviewer 2 Report

Comments and Suggestions for Authors

My comments to the authors are listed below. 

1. I recommend that the authors provide some of the supplemental information (such as the type of tissue used, adjustments for confounding, etc.) in the main manuscript. Considering that the authors used blood, did they attempt to correct for a difference in the cell proportion in their analysis? The analyses by which they derived the dietary and lifestyle inflammatory scores need to be expanded in greater detail. 

2. It is important to discuss how the author's findings fit with existing (if there are) postmortem brain studies assessing the methylation of the KALRN gene in the brain. Are there any postmortem KALRN expression studies specifically in MDD patients? That would support the author's narrative of the importance of KALRN methylation for the association with depression.

3. It is unclear why the authors didn't widen their analysis to the Moli-sani Study sample with existing pyrosequencing data rather than using data from an independent sample. Even though both samples originated from the same cohort, the findings in the smaller N could be confounded in a way that the original sample was not. Additionally, since the authors also tested individual epi-marks, have they corrected for multiple tests in their analysis? This information was not obvious if it was included.

4. Have the authors tried to investigate the correlation between the expression of the KALRN gene and the methylation status cg13549966?

5. The analytical approach of testing for the association between methylation in KALRN and depression is not well developed. It seems that the authors evaluate their findings by incrementally adding additional predictors (covariates). While not necessarily wrong, I wonder why they have not included the model with all predictors (covariates) and compare the full model against the reduced model to evaluate the overall impact of the predictors (covariates) on the association, if that was what was their goal.

6. The rationale for including all CpGs together in the multivariate analysis is not well articulated. Is it for increased power? In this reviewer's opinion, before including all CpGs together in the analysis, have the authors assessed the correlation between the epigenetic marks? In general, it is accepted to include those in a single model if they are correlated.

Comments on the Quality of English Language

the English language is fine. 

Author Response

Dear Editor,

Thank you and both the reviewers for considering our manuscript for publication on the IJMS. Please find below response to reviewer’s comments (in violet).

To highlight the changes throughout the manuscript, we uploaded a tracked changes version of the manuscript in pdf format.

Should you or any of the reviewers need any additional clarification, please do not hesitate to contact us.

Regards,

Alessandro Gialluisi (on behalf of all co-authors)

Reviewer #2:

My comments to the authors are listed below. 

  1. I recommend that the authors provide some of the supplemental information (such as the type of tissue used, adjustments for confounding, etc.) in the main manuscript. Considering that the authors used blood, did they attempt to correct for a difference in the cell proportion in their analysis? The analyses by which they derived the dietary and lifestyle inflammatory scores need to be expanded in greater detail.

We thank the reviewer for the useful suggestion. We added the required details (use of blood samples) to the Materials and Methods section (Subjects and samples paragraph).

Since the methylation analysis was performed on buffy coats, to account for potential confounding by white blood cell populations heterogeneity, we adjusted all our associations for granulocytes, monocytes and lymphocytes fractions. This revealed substantially stable associations, in line with those observed without this adjustment.

Moreover, we would like to highlight that all the necessary details on the construction of the DIS and LIS scores are reported in Supplementary Materials, as we stated in the Methods section (Statistical Analysis paragraph).

  1. It is important to discuss how the author's findings fit with existing (if there are) postmortem brain studies assessing the methylation of the KALRN gene in the brain. Are there any postmortem KALRN expression studies specifically in MDD patients? That would support the author's narrative of the importance of KALRN methylation for the association with depression.

We thank the reviewer for the useful suggestion. After a thorough search of previous literature in the field, we only found evidence of under-expression of KALRN mRNA (Hill et al., 2006, doi: 10.1038/sj.mp.4001792) and of the kalirin protein in postmortem brains of schizophrenia patients (Rubio et al., 2012, doi: 10.1016/j.biopsych.2012.02.006; Deo et al., 2012, doi: 10.1016/j.nbd.2011.11.003). Similar evidence was found for another comorbid condition, Alzheimer’s disease, both at the mRNA and at the protein expression level (Youn et al, 2007, doi: 10.3233/jad-2007-11314; Murray et al., 2012, doi: 10.1016/j.neurobiolaging.2012.02.015). Although we were not able to find a study reporting differences in kalirin expression between depression cases and controls, we found evidence of a decrease of kalirin-7 levels in the hippocampus of male rats exposed to chronic social defeat stress, one of the triggering mechanisms used to model major depression in rodents (Qiao et al., 2014, doi: 10.1016/j.bbr.2014.08.040). Similarly, chronic unpredictable mild stress (CUMS), used to recapitulate depression-like behaviors in rodents and causing a decrease in spine density in the Orbitofrontal Cortex, was accompanied by both depression-like behaviors and decreased expression of kalirin-7 and postsynaptic density protein 95 in this region (Qiao et al., 2016, doi: 10.1155/2016/8056370). We have now added this evidence to the Discussion section.

  1. It is unclear why the authors didn't widen their analysis to the Moli-sani Study sample with existing pyrosequencing data rather than using data from an independent sample.

As we briefly explained in the limitations paragraph, as we carried out the first ad-interim analysis of pyrosequencing data, we realized that despite providing a fine-grained resolution, this did not allow to enlarge the analyzed region, entailing a high cost-efficacy ratio of this analysis. This, along with the new availability of EPIC array data from a parallel project, prompted us to focus on the analysis of CpG array data, rather than carrying out pyrosequencing. On the basis of financial support availability, we plan to complete the analysis of the whole Moli-sani sub-cohort recruited at follow-up (including the samples analyzed so far with pyrosequencing) in the next two years, so to be able to increase the sample size of the analysis.

Even though both samples originated from the same cohort, the findings in the smaller N could be confounded in a way that the original sample was not.

Since the sample size of the two sub-cohorts are quite comparable and we used similar adjustments for both analyses, we see differential confounding of the analyzed relationship highly unlikely. Since the tested CpGs are different, this aspect cannot be assessed with the data currently available. Still, we added this potential critical point – which will be certainly clarified when all the samples will be analyzed through the EPIC array - to the limitation section. Should we not get the point of the objection, we are happy to receive further requests by the reviewer

Additionally, since the authors also tested individual epi-marks, have they corrected for multiple tests in their analysis? This information was not obvious if it was included.

As we mentioned in the caption of Table 2 and in the Results section, we applied a Bonferroni correction for multiple testing to the analysis of CpGs annotated to the KALRN gene which rescued from EPIC array. To make this more explicit and the correction even more robust, we added the following sentences to the Methods section:

Analysis of pyrosequencing data paragraph:

“A Bonferroni Correction for multiple testing was applied as appropriate, based on the number of CpGs and traits analyzed. Specifically, we corrected the α threshold for three CpGs and two traits tested in the analysis of pyrosequencing data (for a total of six independent tests, α = 0.008).”

Analysis of array data paragraph:

“In this analysis, we applied a Bonferroni correction for 48 independent tests (corrected α = 0.001), namely the sum of CpGs retained in the stepwise regression analysis of the MoCA (23 CpGs) and of the PHQ9 score (25 CpGs).”

Furthermore, we modified the Abstract, Results and Discussion sections accordingly.

  1. Have the authors tried to investigate the correlation between the expression of the KALRN gene and the methylation status cg13549966?

We thank the reviewer for the insightful suggestion. We considered carrying out this analysis but at the moment – since we don’t have a wet lab group anymore - we don’t have the necessary expertise and funds to perform this and other functional analyses. Hopefully, the publication of these data may help us finding funds and collaborations to this purpose, as we are already planning functional validation analyses. For this reason, we added the lack of functional validations to the limitations of the present study.

  1. The analytical approach of testing for the association between methylation in KALRN and depression is not well developed. It seems that the authors evaluate their findings by incrementally adding additional predictors (covariates). While not necessarily wrong, I wonder why they have not included the model with all predictors (covariates) and compare the full model against the reduced model to evaluate the overall impact of the predictors (covariates) on the association, if that was what was their goal.

We thank the reviewer for this suggestion, which allows us to clarify the aim of our study. Indeed, our goal was to simply test associations between epigenetic variation in the KALRN gene and interindividual variation in depressive symptoms and cognitive performance, as stated in the Abstract and the last paragraph of the Introduction section. For this reason, we built incrementally adjusted models in the analysis of pyrosequencing data, and – even more – we built multivariable stepwise regression models in the analysis of CpGs from EPIC array. Evaluating the impact of the covariates on cognitive performance and depressive symptoms goes beyond the scope of the study. Indeed, we are working in parallel on global models for the prediction of such traits, including many more predictors than those used here as covariates, through a more comprehensive approach.

  1. The rationale for including all CpGs together in the multivariate analysis is not well articulated. Is it for increased power? In this reviewer's opinion, before including all CpGs together in the analysis, have the authors assessed the correlation between the epigenetic marks? In general, it is accepted to include those in a single model if they are correlated.

We used a stepwise regression model to i) account for the potential concurrent influence of multiple CpGs on the traits of interest and ii) keep under control collinearity biases among CpGs, which may affect associations, resulting in false positives. This way, we selected in the final model only those CpGs associated with a decrease in the Akaike Information Criterion (AIC), namely those entailing a positive tradeoff between goodness of fit and parsimony of the multivariable model, which allowed us to identify associated CpGs through a sensitive but robust approach.

Round 2

Reviewer 2 Report

Comments and Suggestions for Authors

I have no further questions to the authors

Comments on the Quality of English Language

the English language is fine